# Research on Electric Field—Induced Catalysis Using Single—Molecule Electrical Measurement

**DOI:** 10.3390/molecules28134968

**Published:** 2023-06-24

**Authors:** Jieyao Lv, Ruiqin Sun, Qifan Yang, Pengfei Gan, Shiyong Yu, Zhibing Tan

**Affiliations:** College of Chemistry and Chemical Engineering, Inner Mongolia University, Hohhot 010021, China

**Keywords:** electric field catalysis, single-molecule electronic techniques, chemical reactions

## Abstract

The role of catalysis in controlling chemical reactions is crucial. As an important external stimulus regulatory tool, electric field (EF) catalysis enables further possibilities for chemical reaction regulation. To date, the regulation mechanism of electric fields and electrons on chemical reactions has been modeled. The electric field at the single-molecule electronic scale provides a powerful theoretical weapon to explore the dynamics of individual chemical reactions. The combination of electric fields and single-molecule electronic techniques not only uncovers new principles but also results in the regulation of chemical reactions at the single-molecule scale. This perspective focuses on the recent electric field-catalyzed, single-molecule chemical reactions and assembly, and highlights promising outlooks for future work in single-molecule catalysis.

## 1. Introduction

Catalysis can regulate the rate of chemical reactions and selectively synthesize products, which is one of the most important methods of controlling chemical reactions. In addition to using catalysts, external stimuli, including light [1,2,3], heat [4,5], electrochemical stimuli [6,7,8], electric fields [9,10,11] and magnetic fields [12], etc., are also typically used to excite catalytic reactions. Due to the flexible adjustment of parameters, such as field strength and direction, using an electric field to catalyze chemical reactions has unique advantages [13].

In the past two decades, single-molecule measurement techniques and data analysis methods have developed rapidly, allowing users to obtain precise and stable conductance distribution signals [14,15,16]. These techniques were gradually used to explore the physical and chemical properties of chemical changes [17], such as trapping reaction intermediates and transition states [18,19,20] or analyzing the thermodynamic and dynamic properties of a reaction [21,22,23]. As shown in Figure 1, mainstream single-molecule electrical characterization techniques include single-molecular break junction (mechanically controllable break junction (MCBJ) [15,24,25,26], scanning tunneling microscope break junction (STM-BJ) [27,28,29,30,31], graphene–molecule–graphene single-molecule junction (GMG-SMJ) [20,32,33,34], and single-molecule self-assembled monolayer (SAM) technologies [35,36,37,38].

The single-molecule break junction electrical characterization method uses a precise displacement control technique to control the continuous opening and closing of two tiny electrodes, forming a nano-spacer that matches the molecule during the opening of the electrodes, and the anchoring groups on the molecule are connected to the electrodes to form a circuit. Relying on current detection devices with ultra-high sensitivity in the external circuit to monitor the current in real time, a large number of conductance distance curves are obtained and then statistically analyzed to obtain accurate molecular conductance [39]. Many studies have proved that it is not difficult to add a strong electric field to this single-molecule technology [21,25,40]. The advantages of the instrument include technical support for studying the mechanism of nanoscale chemical reactions catalyzed by electric fields, as well as the possibility of the synthesis of complex organic materials and the formation of multifunctional molecular devices.

In this review, we briefly summarized the significant catalytic roles of electric fields on the scale of single-molecule chemical reactions. The catalytic effects of electric fields on the Diels–Alder, cracking, and coupling reactions and the mechanism of the reaction were investigated by relating the electric field intensity [41,42,43] and the electric field direction [17] to the direction of the chemical reaction [44,45]. We also summarize the catalytic role of electronic effects in chemical reactions at the single-molecule scale [46]. Based on the precise regulation of molecules by an electric field, the researchers also realized that electric field induced the molecular assembly [47]. We conclude the article by exploring the prospective challenges of reaction systems for single-molecule catalysis, as well as technical characterization tools.

## 2. Electric Field Catalytic Reactions

### 2.1. Diels–Alder Reactions

The Diels–Alder (DA) reaction is an important type of reaction in organic chemistry, and it is widely used in basic organic synthesis [48], the production of fine chemicals [49], and other fields. The reaction uses a 1,4-cycloaddition reaction of a diene reagent with a conjugated diene to form a six-element cyclic alkene. At room temperature, it is difficult for the reaction to proceed spontaneously, and a Lewis acid is generally used as the catalyst [50]. Charged radicals can also regulate the reaction process, though the reaction is less efficient [51,52,53]. Therefore, a more efficient means of regulation needs to be developed. Shaik and colleagues once predicted that the barrier of some Diels–Alder reactions would be influenced by the external electric field (EEF) [54]. Thus, the EEF holds promise in regulation of DA reaction progression.

In 2016, the Coote and Diez-Perez groups were the first teams to experimentally prove that an electric field could accelerate the progress of non-redox bond formation [17]. As shown in Figure 2A, the researchers measured the conductance of a Diels–Alder reaction between a diene (a furan) and dienophile (a norbornylogous bridge, (±)-NB, tricyclo[4.2.1.0^2,5^]non-7-ene-3,4-dimethanethiol) using a scanning tunneling microscopy break-junction (STM-BJ) technique. The reaction produced four structurally different DA products, each of which had two diastereoisomers with the furan substituent located on the left (see the blue diastereomer in Figure 2B) or right (red diastereomer) side of the molecule. In the negative bias case, the potential for the formation of the red isomer will decrease with increasing field strength; the probability of detecting the reaction product is significantly higher and positively correlated with the applied bias voltage, as shown in Figure 2C. However, in the case of positive bias voltage, the formation potential of the blue isomer is independent of the field strength, and the probability of product generation decreases. Since the dienes are electron-rich and the amphiphiles lack electrons, the negatively biased EEF makes the transition state conform to the lowest potential barrier by stabilizing the charge resonance. 

In addition, the STM break-junction “flicker” experiment showed that the conductance of the molecular bridge is constant under positive voltage bias, as shown in Figure 2D. Conversely, under the negative bias, the product formation increased 4.4-fold from 4.2 to 18.6%. It was found that the molecular configuration was significantly affected by the negative bias through quantum calculation. The negative bias energy significantly reduces the reaction barrier. When an electric field is applied to electrostatically stabilize these covalent compounds, a small charge separates the resonance, resulting in the overall stabilization of the molecule or transition state. This study demonstrated the ability of electric fields to manipulate chemical reactions and the principle for the approach to multiphase catalysis.

Previous studies demonstrated that electric fields affect the rate of the DA reaction and tune the relative orientation between the oriented external electric field (OEEF) and the reaction axis for selective electrostatic catalysis of multistage reactions. Meanwhile, the capture of chemical reaction intermediates helps us to better explain the stepwise pathway of the chemical reaction [17]. In 2021, the Guo group and their collaborators reported the precise temporal trajectory and detailed DA reaction pathway directly observed on in situ label-free graphene-based single-molecule devices using precise single-molecule detection [55]. Chen et al. first designed and synthesized conjugated molecules with maleimide as the functional center and modified with amino groups at the ends, which were attached between graphene point electrodes with carbonyl functional groups via amide bonds to construct stable single-molecule devices. The single-molecule devices were jointly characterized by their self-developed ultra-high spatial and temporal resolution photoelectric integrated detection system with electrical and optical dual-mode methods, providing the first direct experimental evidence of synchronization of the reaction (Figure 3C). They first demonstrated the accepted mechanism of synergistic reactions and captured the charge–transfer complex salts that passed through the corresponding key intermediates before generating products with an endo or exo conformation. Next, by recording the precise time trajectories and detailed reaction paths through a single-molecule electrical detection platform, they revealed a new mechanism for a second stepwise (via ampholytic intermediates) addition reaction (Figure 3A). After clarification of the studied mechanism, the synergistic or stepwise DA reactions were regulated by varying the bias and temperature.

A new mechanism for electric field-catalyzed Diels–Alder addition reactions was revealed by combining experiment and theory: a strong electric field (~109 V/m) was applied, which decreased the potential energy of the key intermediate in the stepwise reaction (the ampholytic intermediate that only adds one bond), thereby greatly enhancing its stability (Figure 3B). This electric field catalytic effect allowed the discovery of the stepwise reaction pathways that had never been observed experimentally in the system, adding to the conventional knowledge of Diels–Alder addition reactions. The researchers also performed a careful analysis of the thermodynamics and kinetics of these processes, which led to the establishment of a new approach to the regulation of intermediate lifetimes, as well as chemical reaction pathways, via electric fields. Mejía et al. employed molecular dynamics (MD) coupled to quantum transport simulations alone to perform theoretical calculations of DA reactions at the single-molecule scale [56]. The calculations show that the DA reaction activation energy is dominated by entropy and that the chemical reactions are thermodynamically influenced. The construction of molecular junctions can also affect the reaction transition states and, thus, modulate the reaction paths of DA reaction occurrence, further providing important theoretical support for single-molecule junctions to monitor antichemical reactions. This observation provides unlimited possibilities for understanding many mechanisms that are difficult to decipher in organic chemistry and lays the foundation for the regulation of chemical reactions and the evolution of the life sciences.

Electric fields can also regulate the direction of the chemical reaction. Hu et al. reported the first example of in situ label-free single-molecule detection of thermally reversible DA reactions using a scanning tunneling microscopy break junction (STM-BJ) technique equipped with a thermocouple mounted under a substrate with feedback control [44]. The thermally reversible DA reaction using anthracene-2,6-diamine (AnAm) as the diene and fullerene C_60_ as the pro-diene is shown in Figure 4A. A homemade STM-BJ coupled with a feedback-controlled thermocouple was used to capture individual molecules (Figure 4B). In Figure 4C, the initial and terminated conductance states of the reaction-based single-molecule junctions can be reversibly switched in situ between two different temperature levels. After testing using different bias voltages, the reaction rate constant (k) calculated via conductance was 1.083 × 10^−2^ min^−1^, which is significantly higher than the k values obtained using UV-vis measurements (7.49 × 10^−3^ min^−1^) for the forward reaction. For the reverse reaction, the two reaction rate constants are similar (5.0 × 10^−5^ and 5.4 × 10^−5^ mM L^−1^ min^−1^). In addition, impressively, quantitative statistical analysis of the single-molecule reaction kinetics showed that the directed external electric field selectively accelerates the forward DA reaction by a factor of more than three, while not affecting the reverse reaction. The theoretical calculations indicated that the applied electric field stabilizes the dipole in the transition state, thus reducing the energy potential barrier for the forward reaction. Thus, the reaction kinetics suggest a significant selective acceleration effect of EEF on the forward direction of this DA reaction. However, the theoretically predicted intermediate state signal was not directly detected via conductance in this work.

Subsequently, the Diels–Alder class of reactions was investigated using single-molecule techniques by Huang et al. [45]. The selective electrostatic catalysis using OEEFs in different processes of the cascade reaction was investigated by tuning the relative orientation between the oriented external electric field (OEEF) and the reaction axis on a single-molecule scale using the MCBJ method. A two-step cascade reaction at room temperature was chosen, in which compound A [3,6-di(4-pyridyl)-1,2,4,5-tetrazine] undergoes an anti-electron-demand Diels–Alder (iEDDA) reaction with dihydrofuran to form b, followed by an aromatization reaction to form c (left column of Figure 5A,B). This Diels–Alder reaction has a reaction axis (a→b) orthogonal to the orientation of the OEEFs, while the subsequent aromatization process (b→c) exhibits a non-orthogonal configuration between the reaction axis and the OEEFs (Figure 5A, top panel), which provides an experimental platform for evaluating electric field-induced selective catalysis of chemical reactions. The reaction kinetics of this two-cascade reaction was determined via single-molecule conductance monitoring using the MCBJ technique; the characteristic conductance peaks for all three molecules are shown on the right side of Figure 5B. The changes in the characteristic conductance peaks in the conductivity statistics plot all visualize the course of the reaction. Experiments at the nanoscale reactor revealed that if the applied electric field is perpendicular to the reaction axis, the electric field has no effect on the chemical reaction; if the electric field has a component in the direction of the reaction axis, the electric field can increase the reaction rate by more than one order of magnitude, and the electric field plays a catalytic role in promoting the reaction. The intermediate state of the chemical reaction pathway was confirmed via single-molecule device electrical transport simulations; the results of transition state calculations showed that the directed electric field can effectively stabilize the transition state of the chemical reaction, thus reducing the reaction energy barrier [57,58,59]. Therefore, the application of electric fields can provide a new opportunity to tune the reaction rate and selectivity of chemical reactions for efficient chemical synthesis and future green chemistry.

### 2.2. Cleavage Reactions

In the 1970s, Pocker et al. pioneered theoretical models showing that electrostatic power can catalyze the formation or breaking of chemical bonds [60,61]. For non-polar covalent bonds, the molecule exhibits a resonantly stable state. When the electric field is present, it changes the stabilization of the covalent bond, reduces the dissociation energy of the molecule, and causes molecular cleavage. Initially, the electric field is provided by the charged functional groups [62,63]. However, the electrostatic effect is more obvious in the gas phase, and it is greatly influenced by the solubility in the liquid phase. Exploring the influence of applied electric fields on molecular cracking is conducive to expanding the research scope of electrostatic catalysis mechanisms, and is of great significance to the regulation and application of electrostatic catalysis.

In 2018, Michelle L. Coote et al. explored the ability of electric fields to rapidly undergo irreversible alkoxyamine cleavage (C-O bond breaking). At room temperature and under different magnitudes of bias stimulation, bridging alkoxyamine molecules between STM gold tips and gold substrates were used to demonstrate that alkoxyamines can undergo irreversible alkoxyamine cleavage (C-O bond breaking) (Figure 6A) [41]. At low deviations (<100 mV), only the parent alkoxylamine molecule (1 × 10^−5^
*G*_0_) is present in the system. However, between 100 and 200 mV, a mixture of nitrogen oxides and parent alkoxylamines is present. Above 200 mV, only nitrogen oxide radicals are detected (Figure 6C,D). Nitrogen oxide has a known affinity for gold surfaces, and the same conductivity characteristics were observed in control experiments using standard nitrogen oxide (4-amino-TEMPO) solutions, confirming that the nitrogen oxide radicals are, indeed, products of OEEF-catalyzed (C-O) alkoxyamine bond breakage (Figure 6B). The role of electric fields in promoting the homolysis of alkoxyamines is explained via quantum chemical calculations. The reaction spectra in the different intensities of electric fields are aligned along the N-O bond axis, which indicates that the homolysis of alkoxyamines can be promoted by the electric field up to 35 kJ mol^−1^. The effect of this energy barrier reduction is consistent with the expected p-nitro radical (N-O• ↔ N^+^•-O^−^) and the stabilization of charge separation resonances, conclusively demonstrating that the electrostatic environment in the solution catalyzes the formation of radicals through cleavage, offering the prospect of electrostatic catalytic decomposition, instead of triggering catalysis.

Recently, the Latha research group systematically quantified chemical rate enhancement using an electric field to clarify the catalytic effect of EEFs [42]. Using the STM-BJ technique, they found that the electric field could catalyze the homogeneous cleavage of the radical initiator 4-(methylthio)benzoic peroxyanhydride in the absence of co-initiators or photochemical activators (Figure 7A). The conductance changed over time, as shown in Figure 7B. It was also demonstrated that the reaction rate in the electric field was affected by the solvent and increased linearly with the dielectric constant of the solvent.

To quantify the effect of bias on molecular catalysis, this research group used the HPLC peak area integral to obtain the curve of the molecular concentration of the reactant and product over time. With a bias imposed at 100 mV, the half-life of 1 is t_1/2_ = 63.0 min, matching the formation rate of 2, thus suggesting that 1 is selectively transformed to 2 as the chemical reaction proceeds. Meanwhile, the chemical reaction rate varies significantly under different biases. At a zero bias, the half-life is increased nearly 6-fold from 100 mV, suggesting that gold also catalyzes the reaction, albeit to a small extent. At a 10 mV bias, the half-life is increased 3-fold compared to the 100 mV half-life. This set of data illustrates the clear bias dependence of the reaction rate, thus proving that the reaction is driven by an electric field. The DFT calculations further prove that the catalytic cleavage is occurring because the applied electric field reduces the dissociation energy of the O-O bond, resulting in a more stable product, and revealing the important role of peroxide and radical dipole moments in reducing the dissociation energy using different solvents and applied magnetic fields.

### 2.3. Coupling Reaction of Transition Metal Complexes

A permanent electric field at the electrode–solution interface has also been shown to affect chemical transformations. Computational studies have shown that the oxidative addition of aryl halides at the palladium center can be affected by an external electric field [64]. Latha’s group used scanning tunneling microscopy break junctions (STM-BJs) to demonstrate for the first time that the reactivity of a kinetically inert transition metal complex can be induced by applying an external electric field to affect the coupling reaction [43]. A large electric field was applied to the molecular solution in the region between the STM tip and the substrate, and a mixture of the nickel (0) olefin complex Ni (COD)(DQ) and the iodinated aromatic compound was placed in 1,2,4-trichlorobenzene (TCB) for conductivity measurement. A new significant peak in the conductance value was observed, indicating the generation of two new molecular species with gold-linked groups in solution (Figure 7D, green traces). In contrast, no conductivity peak was observed in the absence of an applied electric field (Figure 7D, orange traces), indicating that the applied electric field promotes the homogeneous coupling of iodinated aryl groups (4-iodothioanisole). It was shown that only in the presence of such an electric field did the nickel complexes undergo aryl iodide coupling chemistry at room temperature. Bias modulation and solvent selection are both strategies used to control the intensity of the local electric field and, subsequently, the extent of organic transformation. The modulation of organometallic coupling in a nanojunction environment by local electric fields highlights the importance of electric field effects in reaction chemistry, and it provides a new strategy to modulate organometallic reactivity.

Theoretical calculations explain the possible path to the energy regulation of chemical reactions. Firstly, the electric field can influence the potential energy of the transition state of the chemical reaction. Aragonès et al. mentioned that the potential energy of the isomer is affected by negative bias [17]. Yang et al. calculated that the electric field can regulate the ZI lifetime [55]. The reduced transition situation can make the chemical reactions easier to perform. Secondly, we can weaken the chemical bond strength. The calculation results of the Coote team and its collaborators prove that electric fields catalyze the formation of carbocation, rather than carbon radicals [41]. Latha’s group calculated that EEF weakens the O-O bond [42]. The reduction in energy promotes molecular cleavage. Therefore, from the perspective of energy, the electric field catalysis at the single-molecule scale can be successfully realized.

## 3. Electronic Catalysis

For electric field catalysis, giving an external strengthening electric field at both ends of the electrode has an impact on the chemical reaction, such as stabilizing the transition state level [17,45,55] or promoting the formation of stable products [42]. Unlike electric field catalysis, electron catalysis requires a positively charged molecular skeleton as an effective electron acceptor. The molecular skeleton receives electrons injected by the electrode to create the molecular junction [65]. Single-molecule scale electron catalysis then occurs, providing more intuitive evidence of the analysis of the catalytic reaction mechanism.

Electrons are elementary particles with a negative electrical charge. The femtosecond chemistry field suggests that manipulating the charge density of molecules may control chemical reactions [66]. Electrons were shown to initiate chemical reactions by preparing molecules in reactive states or initiating highly selective bond dissociation [67]. Recently, electrons have been shown to reduce the chemical reaction barrier and, subsequently, catalyze chemical reactions [68,69,70]. Thus, electron energy had a significant effect on the chemical reactions. The catalysis of electrons was also experimentally confirmed [46]. With the vigorous development of photoredox catalysis and electromechanical synthesis, researchers gained a deeper understanding of the role of electrons in chemical reactions. In 2014, Prof. Dennis P. Curran and Armido Studer proposed the concept of “electron catalysis” [68]. In chemical reactions, electrons can act as a catalyst, reducing a part of the substrate into radical ions, reducing the reaction energy barrier, and promoting the rapid breaking and formation of covalent bonds. The resulting intermediates can spontaneously release electrons, generate products, complete the catalytic cycle, and speed up the reaction rate [68,70].

Study of single-molecule electron catalysis was first conducted by Chen et al. They found that the positively charged molecular skeleton can serve as an effective electron acceptor. The molecular skeleton receives the electrons injected by the electrode, which induces the molecular junction to undergo redox reaction. A single-molecule redox switch was constructed accordingly [65]. Based on an abnormal phenomenon of the cyclic voltammetry test, Chen et al. found that electrons can even serve as effective catalysts for the free radical reaction to induce the dehydrogenation reaction [46]. This reaction is shown in Figure 8A. The traditional theory holds that the 1,2-di(4-pyridinium)ethane (DPA^2+^) skeleton of the conjugated broken ring is not conducive to electron transport, meaning that the conductance will be much lower than the conjugated 1,2-di(4-pyridinium)ethene (DPE^2+^) skeleton. However, the saturated DPA^2+^ backbone exhibits conductance characteristics very similar to those of the conjugated bispyridine–ethylene DPE^2+^ backbone molecules. Therefore, the authors concluded that the conversion of ethane to ethylene occurred in the molecular junction, thereby increasing the conductance. After reduction in the −0.80 V potential, the UV-Vis absorption profile of the saturated DPA^2+^ skeleton was consistent with the DPE^••^, as well as partially oxidized to DPE^•+^ in the single free ground state. Strong electron paramagnetic resonance (EPR) spectrum analysis also captured evidence of DPE^•+^ intermediates obtained via DPA^2+^ reduction. The electron transfer process was manifested as i–vi in Figure 8B. These metastable radicals demonstrate that the DPA^2+^ unit is subjected to an electron-triggered redox process and that the local electric field promotes the dehydrogenation process. They further synthesized two-channel asymmetric ring phanes (Figure 8C) in order to exclude the possibility of an ethylene configuration. Using one channel including an viologen (ExBIPY^2+^) unit and the another that included a DPA^2+^ or DPE^2+^ unit, rigid structure molecular skeletons (3-DA·4PF_6_ and 4-DA·4PF_6_) were synthesized. In Figure 8D, the similar conductance of both models proved that no isomerization occurred and reconfirmed the DPA^2+^-to-DPE^2+^ dehydrogenation process. The present study demonstrated the mechanism of electron catalysis at the single-molecule scale and provides a new means of investigating the mechanism of electrocatalytic hydrogen production in depth.

## 4. Electric Field Catalytic Assembly

The molecular assembly provides a more efficient solution for achieving a wider range of materials and devices. The molecular assembly can be achieved through two main pathways: self-assembly and auxiliary assembly. Among these methods, electric fields are expected to play an important role in catalytic assembly.

In 2020, the Hong research group applied electric fields to realize the induced assembly of single-molecules, which increased the technical possibility of the future realization of molecular transistors [47]. As shown in Figure 9A, they first measured a single triphenyl molecule using the STM-BJ technique. At a 0.1 V bias, the tribiphenyl high conductance was 10^−3.56^*G*_0_. When they increased the bias of the two segments of the electrode from 0.1 V to 3.5 V, the high conductance was almost unchanged (10^−3.36^*G*_0_), and the clear low conductance peak for 10^−5.30^*G*_0_ appeared (Figure 9B). The conductance change stemmed from the transformation of the molecular junction in the electric field-induced circuit from single-molecules to molecular assemblies. Combined with theoretical calculations, they found two reasons for assembly. One reason is that the electric field reduces the rotation of the benzene ring bond, meaning that the conjugate structure of triphenyl is enhanced, and the other reason is that the electric field reduces the stroke potential energy. The experiments showed that the electric field can also achieve induced assembly for molecules of three other resistance and anchor groups, as shown in Figure 9C,D. This work provides important insights into the mechanism of intermolecular transport at the nanoscale using electric fields, providing possibilities for monitoring and control of the molecular assembly.

## 5. Conclusions and Outlook

Electric field catalysis shows various potential applications in single-molecule-based chemical reactions, especially in the study of chemical reaction processes. In this paper, important research results in the field of physicochemical research in recent years are reviewed from the perspective of single-molecule chemical reactions catalyzed by electric fields. Firstly, the monitoring of single-molecule charge transport properties and quantum behavior under electric fields is presented. We focus on the modulation of single-molecule charge transport by electric fields for different types of chemical reactions and various modulations of single-molecule chemical reaction mechanisms, including Diels–Alder reactions [17,44,45,55], cleavage reactions [41,42], and coupling reactions [43]. Next, the monitoring of single-molecule redox reaction processes in electron effects is presented to explore the effect of electron energy on chemical reactions [46]. Finally, the intermolecular transport mechanism of single-molecule electric field-catalyzed assembly is presented, for which monitoring and controlling the molecular assembly offers the possibility of further use [47].

An electric field is applied at both ends of the electrodes of the single-molecule device. As the electrode spacing decreases, the electric field acting on the molecules on both sides of the electrode increases significantly, and the nanoscale electrodes generate strong electric fields in single-molecule devices [71]. Therefore, using the physical properties of molecules and state control of chemical reactions, applying smaller voltages at both ends of the nanoscale electrodes can lead to more sensitive and efficient, as well as strong, electric field single-molecule electric field catalytic devices [72]. As of now, study in this field has achieved a series of important results [73]. However, studies of electric field catalytic processes of single-molecules, especially those related to the internal mechanisms and regulation of different chemical reaction types, are still insufficient. There are still many challenges in the improvement of monitoring techniques, the expansion of research systems, and the wide application of research results.

Firstly, we can expand the test to more types of chemical reactions. In addition to DA reaction, many typical chemical reactions can be catalyzed by electric fields, such as the Menshutkin reaction [11], ring opening reactions [74], electrophilic aromatic substitution reactions [75], oxidative addition reactions between palladium catalysts and alkyl/aryl electrophiles [64], and the Kemp elimination reaction [76]. Focusing the units of the chemical reaction on the single-molecule scale has the opportunity to more accurately study the mechanism via which the electric field affects the chemical reaction.

Secondly, we need to develop optical and electrical signal co-characterization equipment. As a precise technical method, ultrafast spectroscopy technology uses pump-detection technology to provide a femtosecond-level optical signal [77,78,79]. The combination of ultrafast spectroscopy and single-molecule electrical measurement instruments enables real-time capture of conductance and photocurrent signals during chemical reactions. After the conductivity signal confirms the capture of the molecule, it is immediately switched to observe the transfer of electrons during the chemical reaction using the photocurrent signal, which significantly improves the accuracy and precision of the test. The technology is expected to obtain the chemical reaction dynamics information at both extreme time scale and extreme spatial scale, as well as analyze the chemical reaction mechanism from multiple dimensions. Despite significant challenges, in-depth research in this field can enable significant progress in the physical and chemical fields at the single-molecule level.

## Figures and Tables

**Figure 1 molecules-28-04968-f001:**
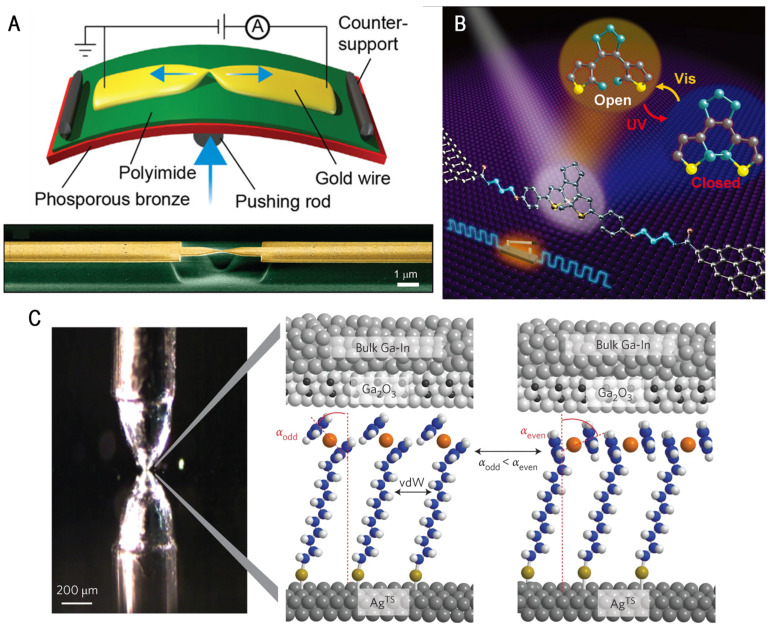
(**A**) Schematic representation of the MCBJ technology and scanning electron microscopy images of wires. Reproduced with permission from [26]. Copyrights, 2018 American Chemical Society. (**B**) Schematic of graphene–diarylethene–graphene junction. Reproduced with permission from [34]. Copyrights, 2016 American Association for the Advancement of Science. (**C**) Schematic illustration of junctions of the EGaIn technology and the mechanism of charge transport across them. Reproduced with permission from [38]. Copyrights, 2013 Nature Publishing Group.

**Figure 2 molecules-28-04968-f002:**
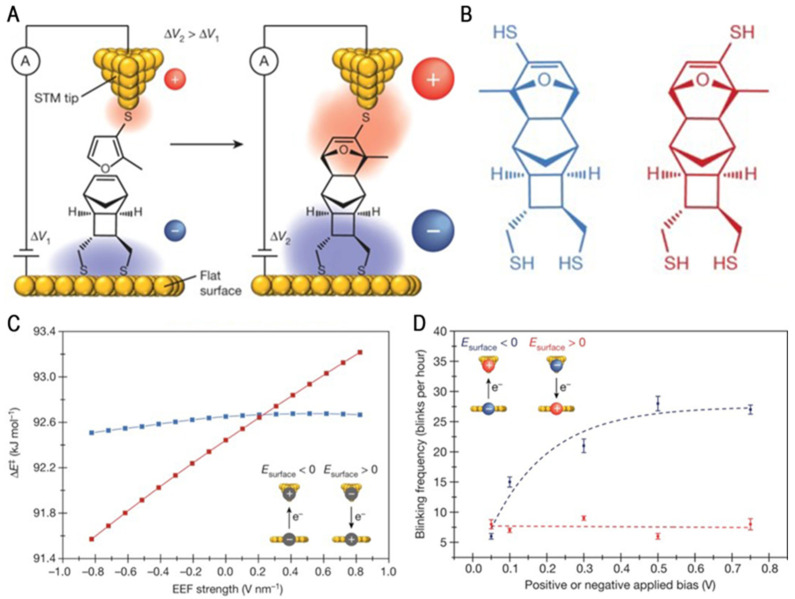
(**A**) The schematic of the DA reaction in STM-BJ conductance measurements. (**B**) The two diastereoisomers of the *exo-syn* product of this reaction. Different positions of the substituent of the furan are shown. (**C**) The predicted effects of the strength and direction of the external electric field (EEF) on the reaction–barrier height (ΔE^‡^) for molecules in B. (**D**) The frequency of blinks (junctions) as a function of the applied bias. Reproduced with permission from [17]. Copyright, 2016 Macmillan Publishers Limited. All rights reserved.

**Figure 3 molecules-28-04968-f003:**
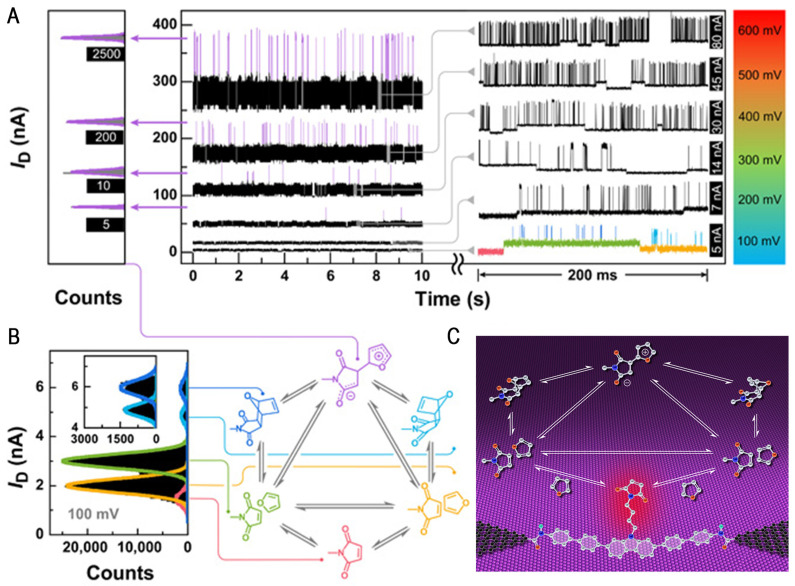
(**A**) Bias voltage-dependent experiments. The left side is the statistical histogram of the zwitterionic intermediate (ZI). The right side is the zoom-in picture of the concerted reaction process. (**B**) Statistical histograms and the corresponding attribution of the six conductance states obtained from Gaussian fittings of I-t measurements. (**C**) The schematic of the single-molecule electrical monitoring platform. Reproduced with permission from [55]. Copyright, 2021 the authors, some rights reserved; exclusive licensee is the American Association for the Advancement of Science. No claim to original U.S. Government works. Distributed under a Creative Commons Attribution Non-Commercial License 4.0 (CC BY-NC).

**Figure 4 molecules-28-04968-f004:**
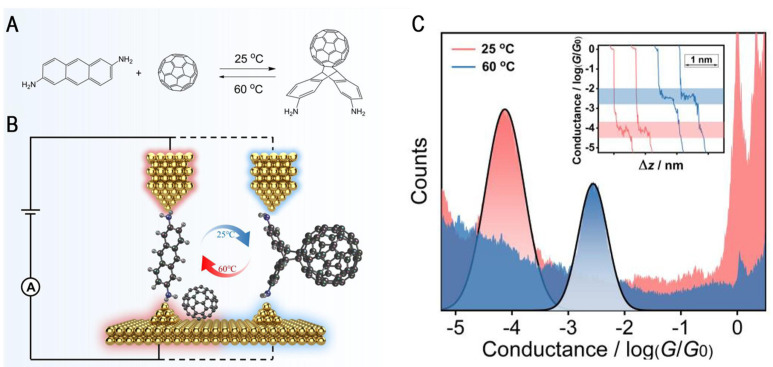
(**A**) The thermal-reversible DA reactions between AnAm and C_60_. (**B**) The schematic of DA process using STM-BJ technology. (**C**) The single-molecule conductance of each compound was characterized through conductance histograms. The inset shows typical traces of both molecule junctions. Reproduced with permission from [44]. Copyright, 2022 Elsevier B.V. All rights reserved.

**Figure 5 molecules-28-04968-f005:**
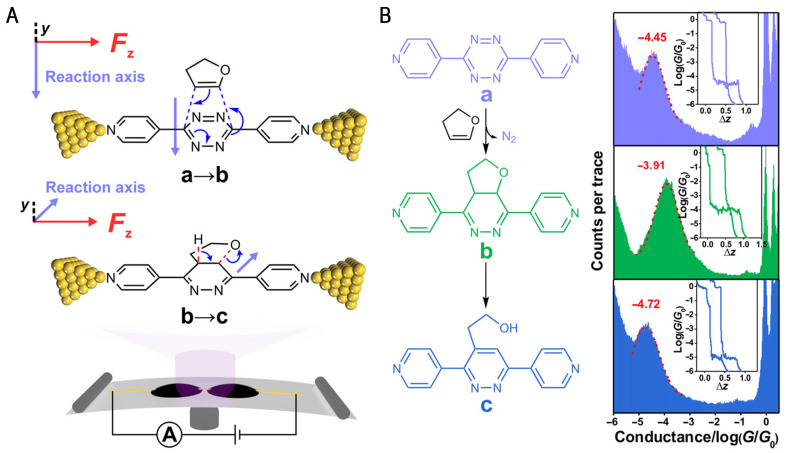
(**A**) The schematic of the MCBJ technique for in situ single-molecule conductance measurement. (**B**) The left column shows the reaction between 3,6-di(4-pyridyl)-1,2,4,5-tetrazine (a) and 2,3-dihydrofuran to form compound b, which goes through an aromatization process to form compound c. The 1D conductance histogram results constructed for the single-molecule conductance of each compound are shown in the right column. The inset shows the typical traces of the compounds. Figure reproduced from reference [45]. Copyright, 2019 the authors, some rights reserved; exclusive licensee is the American Association for the Advancement of Science. No claim made to original U.S. Government works. Distributed under a Creative Commons Attribution Non-Commercial License 4.0 (CC BY-NC).

**Figure 6 molecules-28-04968-f006:**
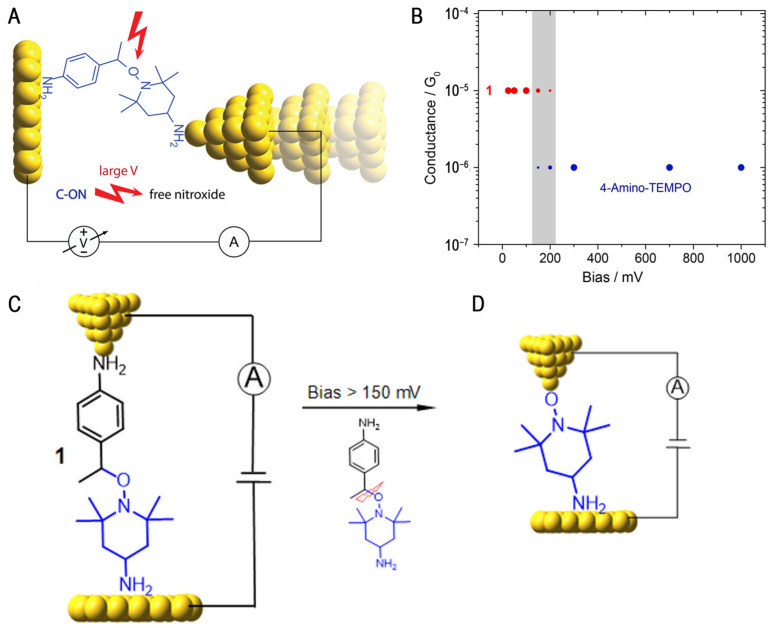
(**A**) The schematic of C-O bond cleavage catalysis by an EEF. (**B**) Molecular conductance changed during electrostatic catalysis in the homolysis of alkoxyamines. (**C**,**D**) A schematic depiction of the STM-BJ setup for a single-molecule junction experiment used to investigate the effect of an external electrical field on the breaking of a C-ON bond. Reproduced with permission from [41]. Copyright, 2017 American Chemical Society.

**Figure 7 molecules-28-04968-f007:**
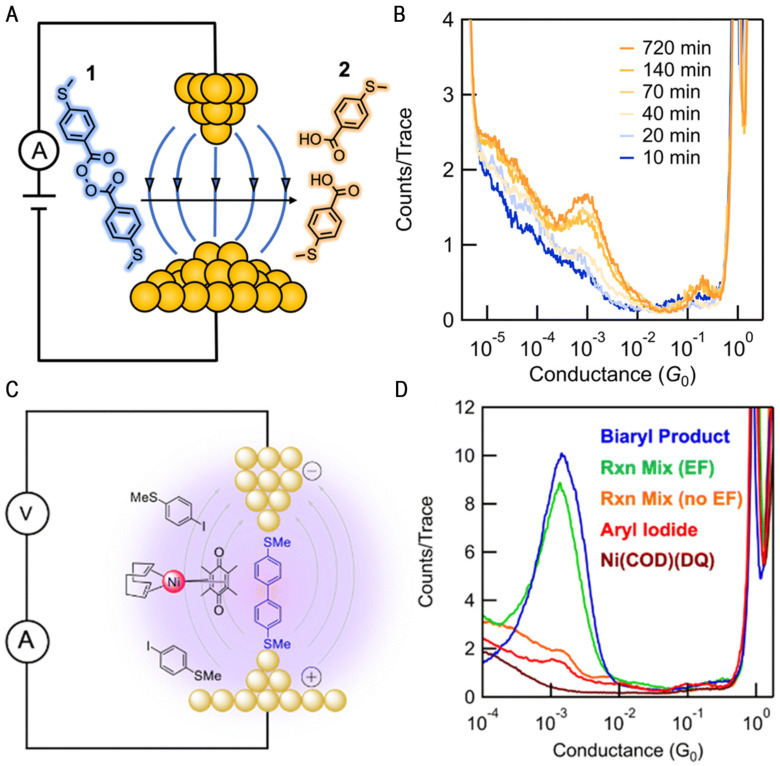
(**A**) The schematic of the homolysis of 4-(methylthio)benzoic peroxyanhydride 1 used to obtain 4-(methylthio)benzoic acid 2. (**B**) The 1D conductance histogram traces measured at different reaction times. Reproduced with permission from [42]. Copyright, 2023 the author. Published by the Royal Society of Chemistry. (**C**) A schematic illustration of the STM-BJ environment. (**D**) Logarithmically binned 1D conductance histograms stemming from traces collected for the coupling reaction of the kinetically inert transition metal complex. Reproduced with permission from [43]. Copyright, 2022 The Royal Society of Chemistry.

**Figure 8 molecules-28-04968-f008:**
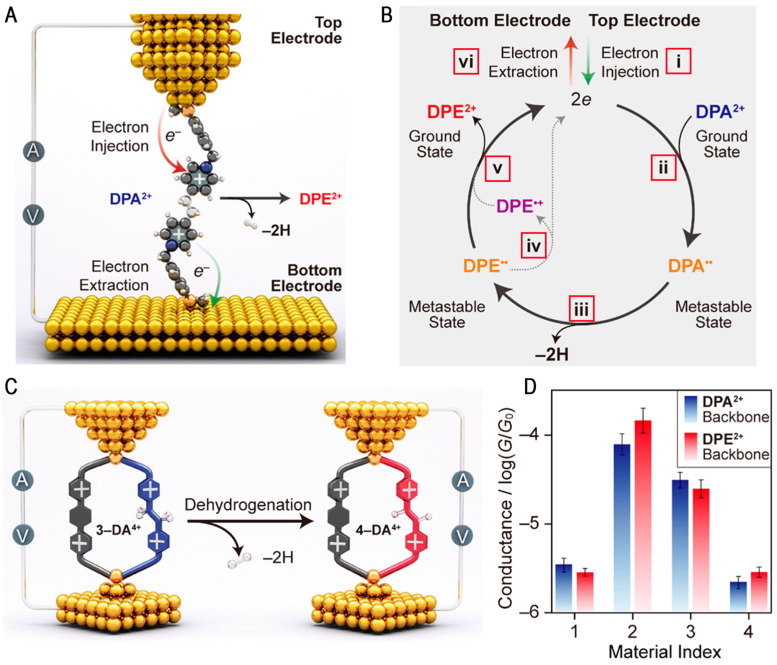
(**A**) The schematic of the electron-catalyzed dehydrogenation at an STM junction. (**B**) A plausible mechanism for the electron-catalyzed DPA^2+^-to-DPE^2+^ dehydrogenation. (**C**) The schematic of the electron-catalyzed dehydrogenation of cyclophanes. (**D**) Conductance comparisons between molecular pairs bearing either a DPA^2+^ (blue) or a DPE^2+^ (red) backbone. Reproduced with permission from [46]. Copyright, 2021 American Chemical Society.

**Figure 9 molecules-28-04968-f009:**
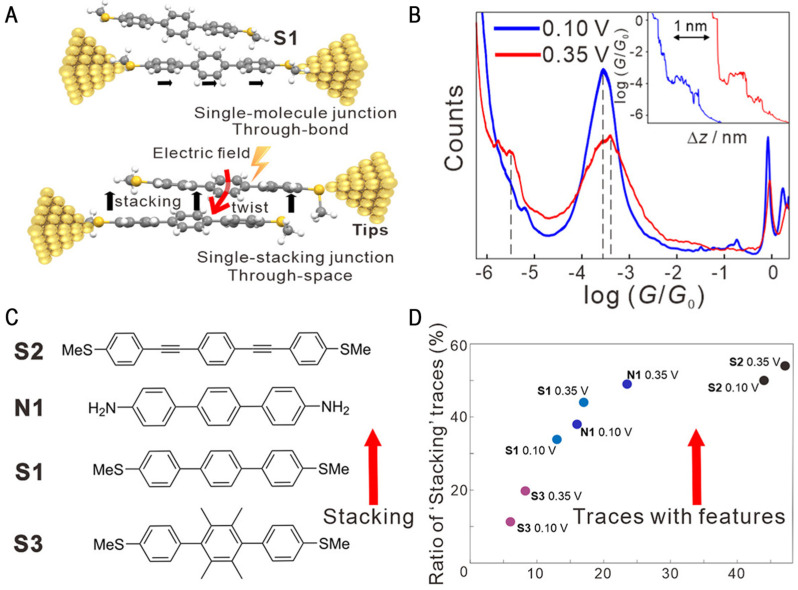
(**A**) The schematic of the EEF-induced single-stacking junctions of terphenyl. (**B**) The 1D conductance histograms under 0.10 (blue) and 0.35 V (red). The inset shows traces of the typical conductance displacement. (**C**) The structures of the molecules used in the formation of single-stacking junctions. The red arrow shows the increase in the formation probability of single-stacking junctions. (**D**) A scatter plot of the ratio of traces with twist and stacking features with the four molecules in C. Reproduced with permission from [47]. Copyright, 2020 American Chemical Society.

## Data Availability

Not applicable.

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
