# Peer review of "Research on Electric Field—Induced Catalysis Using Single—Molecule Electrical Measurement"

_molecules, 2023, doi:10.3390/molecules28134968_

Round 1

Reviewer 1 Report

Yu and Tan reported a review entitled “research on electric field-induced catalysis using single-molecule electrical measurement”. Electrochemistry for catalysis is nowadays an on-the-edge field of research that is gaining more and more interest. The subject treated by this review is highly interesting. However, many points have to be clarified. 

Albeit the abstract seems concise and direct, the introduction is badly written. The syntax is unproperly arranged (this point is constant throughout the whole text) and it is unclear what will be treated through the review. References and subfields of electrochemistry are merged together, often leading to confused statements. The further paragraphs are unbalanced. Half of the review is treating the “electric field catalytic reactions” and mostly focuses on the Diels-Alder transformation. I suggest to change the name of this paragraph to something more specific (and therefore also the title of the manuscript). I cannot see as well a difference between the points treated in this paragraph and the next one, entitled “electronic catalysis”. “Conclusion and outlook” shows again a bit confusing lists of arguments merged together with a logic that is sometimes hard to get. 

As stated above, I believe this review is treating a very interesting field for the research community. However I think that an overall organization as well as a proper re-writing is needed. Thus I suggest the present manuscript for publication in Molecules upon major revisions. 

See comments for authors sections

Reviewer 2 Report

The authors of this article reviewed the regulation of chemical catalysis reaction with an external electric field (EF) at a single molecule level. The single-molecule electronics show a  promising theoretical tool to explore the dynamics of individual chemical reactions. This manuscript provides a details discussion of the recent findings of chemical reactions especially the DA mechanism while modulating with the electric field. To me, this is a good review. The manuscript is well-organized and constructive. However, I feel the author needs to clarify the following points before publication.

1. The author should include the break junction research work by Franco et.al. Please take a look at this article J. Phys. Chem. C 2021, 125, 27, 14599–14606.

2. Tuning chemical reactivity with external stimuli is a hot topic in chemistry, physics, chemical biology and beyond. I wonder within single molecule study can the author provide some picture of coherence? like can you control reaction dynamics from coherent to incoherent regime and vice versa.

3. Tuning an electric field can modify the potential energy landscape, so does the fate of the reaction. Maybe it would be worth including some comments regarding this. 

4. As the author mentioned about the PCET, I am curious what would be major challenges with this type of mechanism to study? Can single molecule study able to control the sequential vs consecutive PCET mechanism. 

Round 2

Reviewer 1 Report

After considering the detailed point-by-point answers and the exhaustivity of the arguments, I therefore suggest the present modified manuscript for publication as it is, without further changes.